# Advance Care Planning in Neurodegenerative Disorders: A Scoping Review

**DOI:** 10.3390/ijerph19020803

**Published:** 2022-01-12

**Authors:** Andrea Giordano, Ludovica De Panfilis, Marta Perin, Laura Servidio, Marta Cascioli, Maria Grazia Grasso, Alessandra Lugaresi, Eugenio Pucci, Simone Veronese, Alessandra Solari

**Affiliations:** 1Unit of Neuroepidemiology, Fondazione IRCCS Istituto Neurologico Carlo Besta, 20133 Milan, Italy; andrea.giordano@istituto-besta.it (A.G.); l.servidio@campus.unimib.it (L.S.); 2Bioethics Unit, Azienda USL-IRCCS di Reggio Emilia, 42100 Reggio Emilia, Italy; ludovica.depanfilis@ausl.re.it (L.D.P.); marta.perin@ausl.re.it (M.P.); 3PhD Program in Clinical and Experimental Medicine, University of Modena and Reggio Emilia, 41100 Modena, Italy; 4Hospice ‘La Torre sul Colle’, Azienda USL Umbria 2, 06049 Spoleto, Italy; marta.cascioli@uslumbria2.it; 5Multiple Sclerosis Unit, IRCCS S. Lucia Foundation, 00179 Rome, Italy; mg.grasso@hsantalucia.it; 6Dipartimento di Scienze Biomediche e Neuromotorie, Università di Bologna, 40126 Bologna, Italy; alessandra.lugaresi2@unibo.it; 7IRCCS Istituto delle Scienze Neurologiche di Bologna, 40139 Bologna, Italy; 8UOC Neurologia, ASUR Marche-AV4, 63900 Fermo, Italy; eugenio.pucci@sanita.marche.it; 9Fondazione FARO, 10133 Turin, Italy; simone.veronese@fondazionefaro.it

**Keywords:** advance care planning, review literature, palliative care, end of life care, ethics, nervous system diseases, neurodegenerative disorders, shared decision making, dementia

## Abstract

Advance care planning (ACP) is increasingly acknowledged as a key step to enable patients to define their goals/preferences for future medical care, together with their carers and health professionals. We aimed to map the evidence on ACP in neurodegenerative disorders. We conducted a scoping review by searching PubMed (inception-December 28, 2020) in addition to trial, review, and dissertation registers. From 9367 records, we included 53 studies, mostly conducted in Europe (45%) and US-Canada (41%), within the last five years. Twenty-six percent of studies were qualitative, followed by observational (21%), reviews (19%), randomized controlled trials (RCTs, 19%), quasi-experimental (11%), and mixed-methods (4%). Two-thirds of studies addressed dementia, followed by amyotrophic lateral sclerosis (13%), and brain tumors (9%). The RCT interventions (all in dementia) consisted of educational programs, facilitated discussions, or videos for patients and/or carers. In conclusion, more research is needed to investigate barriers and facilitators of ACP uptake, as well as to develop/test interventions in almost all the neurodegenerative disorders. A common set of outcome measures targeting each discrete ACP behavior, and validated across the different diseases and cultures is also needed.

## 1. Introduction

High-quality and comprehensive care is a challenge for health systems around the world [1]. Modern medicine is often characterized by multiple choices brought about by technological advances and a multitude of options of treatments. In a complex healthcare landscape characterized by ageing, chronic diseases, and severe illnesses, it is crucial to promote a personalized care approach, based on the incorporation of patient values, personal ethical principles, preferences, and goals of care into present and future individual clinical decisions. The goal of this approach is to align evidence-based practice and person-centered care focusing on the decision-making process [2].

While the alignment of treatment with patient needs is the core element of shared decision-making [3] during the whole care pathway, the goal of Advance Care Planning (ACP) is to help persons in making treatment decisions in advance. According to the most recent definition of ACP, it ‘enables individuals who have decisional capacity to identify their values, to reflect upon the meanings and consequences of serious illness scenarios, to define goals and preferences for future medical treatment and care, and to discuss these with family and health-care providers’ [4]. Moreover, ‘ACP addresses individuals’ concerns across the physical, psychological, social, and spiritual domains’. This broad definition of ACP is in line with the idea that the decisions near to the end of life (EOL) are complex, uncertain, emotionally laden, and can change rapidly with changes in clinical conditions [5]. Further, ethnic differences were observed in the concept of autonomy, communication of prognosis, decision-making models, and attitudes toward EOL care. All these factors have implications for ACP [6].

ACP is a communication process [7,8] with a holistic approach to facilitate decision-making. ACP is not a one-time exercise [9]; it requires time and a gradual approach because it includes multiple steps: discussing and exploring personal values, translating them in advance decisions on treatments, choosing a trustee, and documenting advance directives (ADs) through written documents [10]. A specific ACP activity goes beyond written ADs [11]. It is firstly a conversation about values, then about life, death, and dying [12].

Healthcare professionals (HPs) have to develop specific relational and communicative skills to talk about future deterioration and death with their patients, and to best support the whole process [13,14]. In 2019, a systematic review [15] identified 34 interventions to support clinicians in the daily ACP process by providing guidance to the structure and content of ACP conversations. The ethical principle of patient autonomy is the starting point in the interventions examined, but principles of communication and relational ethics had been introduced, focusing on a deeper conversation between the patient and HPs [15].

According to the most recent literature, ACP can improve the quality of patient-clinician communication and EOL care [7], raise the completion of ADs [16], and increase palliative care use [17]; nevertheless, its frequency of use remains low in clinical practice and several barriers emerge. From the patient’s point of view, the unpredictable course of disease, the insufficient knowledge, the hesitancy to discuss personal preferences, and the expectations from doctors represent the most significant barriers; instead, the professional factors hindering ACP are the hesitancy to discuss possible future deterioration with patients, the fear of taking away hope, the lack of knowledge and skills and confidence, and time constraints [7,18,19].

Carrying out effective ACP in patients with neurodegenerative disorders can be even more challenging due to the complex and specific physical, cognitive, psychological needs that those patients have [20]. For example, literature highlights many concerns affecting HPs’ opinions about timely EOL shared decision-making: the identification of an appropriate moment to initiate ACP is still missing, especially in people with dementia [21]. Moreover, clinicians are frequently uncomfortable and lack communication skills for ACP discussions, for example with young patients with a neurological disease, such as multiple sclerosis (MS) [22]. Lastly, while a conceptual scheme for ACP in cancer patients has been developed [23], there is no single theoretical understanding of the contexts or circumstances in which ACP is relevant to people with neurodegenerative disorders, due to the huge differences in the characteristics of the disorders.

The aim of this scoping review was twofold: (1) to map the existing literature on ACP in neurodegenerative disorders targeting patients, carers, and HPs; and (2) to summarize the findings to promote future research and inform clinical practice.

## 2. Methods

Following the Arksey and O’Malley framework [24,25], phases of the scoping review encompassed the formulation of the research question; identification and selection of the relevant studies; data charting, collating and summarizing; and reporting results. We followed the PRISMA-ScR checklist for scoping review conduct and reporting (File S1).

Our main research question was: What is the stage of research concerning ACP in neurodegenerative disorders?

We were interested in studies on ACP interventions (e.g., programs, conversation guides, etc.) targeting patients, carers, HPs, or any combination.

We developed a search strategy for PubMed, and adapted it for Cochrane Database of Systematic Reviews, Cochrane Central Register of Controlled Trials, and PROSPERO registry (for systematic review protocols) from inception to 28.12.2020; trial and dissertation registries (https://clinicaltrials.gov/ct2/home; http://apps.who.int/trialsearch, accessed on 6 January 2022) for unpublished or ongoing studies; previous systematic reviews for additional studies; reference lists of selected studies. Google (selective search: advance care planning guide) was also used to identify other documents and relevant grey literature. The search strategy is reported in File S2.

Two pairs of reviewers pilot tested ten articles to refine the eligibility criteria. Next, the results were split, and the same four reviewers screened titles and abstracts of studies for eligibility. Full-text of the selected studies was then reviewed independently by the reviewers. We included studies if they reported primary qualitative/quantitative research, or secondary research concerned with ACP interventions in neurodegenerative disorders (e.g., amyotrophic lateral sclerosis [ALS], motor neuron disease, dementia, MS, Parkinson’s disease). We excluded studies in which ACP was only part of a more complex intervention, and editorials/narrative reviews unless they presented original study findings. Two reviewers extracted data (checked by a third reviewer) from each included study using an ad hoc electronic form. Data related to study author, methods, country, ACP definition, participants’ characteristics, aims, main results, and funding were extracted. 

Following McMahan et al. (2020) [26] we categorized the interventions as training programs for HPs, videos, educational programs, facilitated discussions, and written-only materials for patients and/or carers. Further, outcomes were categorized using the standardized ACP Outcomes Framework [27] into five domains: process (e.g., behavior change), action (e.g., communication and documentation), quality of care (e.g., goal concordant care and satisfaction), health status, and healthcare utilization. Results were defined as positive if there were significant differences between groups (*p* < 0.05).

Finally, quality of the evidence of the included studies was appraised using different tools according to the different study designs. The (systematic) reviews were assessed using the revised Measurement Tool to Assess Systematic Reviews [28]. As the tool is focused on systematic reviews of randomized controlled trials (RCTs), three questions were omitted and the questions about risk of bias were further adapted. 

For the RCTs, we used the Cochrane tool for risk of bias [29]. 

We used the Critical Appraisal Skills Program [30] tools adapting the different tools. For cross-sectional studies with control groups, we used the relevant case control study checklist. Cohort studies including observational, cross-sectional/retrospective studies, and studies without control groups were assessed using the relevant cohort study checklist. Scores ranged from 0 to 8 points with higher values indicating better quality. 

Finally, we used the Mixed Methods Appraisal tool for mixed-methods studies [31], with values ranging from 0 (no criterion met) to 100 (all criteria met).

Two reviewers evaluated studies separately according to the relevant tool requirements and discussed with a third reviewer possible discrepancies of assessments afterwards. Any discrepancy between the two reviewers in all the stages above was resolved by consensus.

Data were synthesized descriptively to map different aspects of the literature as outlined in our key question. Results of the review are presented in a narrative form, and reported by disease and within dementia by considering the different phases of the ACP process (i.e., ACP introduction; ACP facilitation and barriers/challenges, and ACP documentation).

## 3. Results

Of 9367 references identified, and after initial screening, another 4606 citations were excluded. Of the 263 full-text articles retained for further screening, 216 were excluded because they were not focused on ACP, or ACP was only part of a more complex intervention, had wrong patient population, study design, were protocols, duplicates, or with no full-text available. Fifty-three studies were included (Appendix A; Figure 1) mostly conducted in Europe (24/53; 45%) and US-Canada (22/53; 41%). Among European countries, figures were higher for UK (9/53), Belgium (6/53), and the Netherlands (6/53). Among US-Canada, 19/53 (36%) studies were conducted in the US. Of those, the vast majority (46/53; 87%) was published in the last 5 years (Appendix A). 

Overall, 14/53 (26%) of the studies were qualitative, followed by observational (21%), reviews (19%), RCTs (19%), quasi-experimental (11%), and mixed-methods (4%). Thirty-five of 53 (66%) of the studies were conducted in dementia, followed by ALS/motor neuron disease (13%), brain tumors (9%), Parkinson’s disease (4%), mixed populations (4%), Duchenne muscular dystrophy (2%), and MS (2%). Figure 2 shows the distribution of studies across all the neurodegenerative disorders by study design. 

Twelve (23%) of the studies addressed patients, 10 HPs (19%), 8 carers (15%), or a combination (43%). Of the 43 relevant studies, 20 (38%) were conducted in the outpatient setting, with the others being conducted in nursing homes (21%), community (17%), inpatient setting (4%), or a combination (17%).

The majority of the included studies (34/53, 64%) referred to ACP as a process, process/ADs, process/discussions; six (11%) as EOL discussion/conversation/planning; six (11%) as ADs, ADs/discussion, discussions/directive, discussions/conversations, conversations, written document/ADs; one as goals of care; and one as ‘modifiable factor associated with better outcome’. Five studies (10%) did not report any definition of ACP.

Most of the included studies (83%) did report the source of funding. Seventeen of 44 (38%) were funded by a charity; 16 (37%) by a national institution (government or university); three (7%) by either national institution or a charity; and one (2%) by the European commission; seven (16%) reported no financial support.

### 3.1. Study Quality

Results of the quality assessments are reported in Appendix A and in Appendix A. For the reviews, quality ratings ranged from 5/13 to 10/13; for case-control studies (i.e., intervention studies with control group) from 3/8 to 5/8; for cohort studies (i.e., observational, cross-sectional/retrospective studies, and studies without control groups) from 2/8 to 7/8; for qualitative studies from 6/9 to 8/9, and for mixed-methods studies from 5/7 to 6/7. 

Two of the 10 RCTs were of high methodological quality [32,33]. All the remaining studies had at least some risk of bias (Appendix A). Sequence generation was adequate in eight studies [32,33,34,35,36,37,38,39], and unclear in two studies [40,41]. Allocation concealment was adequate in three studies [33,37,39], inadequate in one [35], and unclear in six studies [34,36,37,38,40,41]. Three studies performed blinding of participants [32,33,36]. Blinding of assessors was performed adequately in seven studies [32,33,36,37,38,39,40], was unclear in two studies [34,35], and inadequate in one [41]. Attrition bias was present in one study [34]. All studies but one had adequate reporting [38]. Finally, in three studies [32,35,41], there was imbalance between groups at baseline.

### 3.2. Dementia

In their recent umbrella review (overview of systematic reviews) Wendrich-van Dael et al. (2020) [42] included 19 reviews and 11 primary articles. ACP was found to be associated with increased completion of ACP documents, increased concordance between care received and prior wishes, and decreased hospitalizations. Six themes emerged on how patients with dementia viewed ACP: timing to the needs of people with dementia and tailoring the approach;varying capacity and readiness to engage in ACP;roles and responsibilities of HPs;impact of relationships on ACP;training;resources needed.

An overarching feature is the diminishing decision-making capacity over time.

Results from the studies reported below are described considering the different phases of the ACP process (i.e., ACP introduction; ACP facilitation and barriers/challenges; and ACP documentation).

#### 3.2.1. ACP Introduction

All the studies reported here were conducted in early-stage/early-onset-dementia.

In their integrative literature review, Geshell et al. (2019) [43] investigated the experiences and perspectives of people with dementia on ACP. They included 18 studies, focusing on engagement in and correlates of ACP participation, ability and attitudes toward ACP participation, and preferences and values for EOL care. Patients generally had a neutral or negative attitude towards ACP, at the same time emphasizing the importance of family involvement when discussing EOL issues. 

By focusing on patients living in the community, Lai et al. (2019) [44] identified key facilitating factors for them to engage with the decision-making process: valuing decision making for the future;timing of initiating conversations;understanding and knowledge of dementia and decision-making for the future;HPs’ communication skills;quality of the relationship;orientation to the future.

Sussman et al. (2020) [45] investigated the patients’ and caregivers’ perceptions of and experiences with ACP; their worries related to future care; and ways supporting positive engagement with ACP. Both patients and caregivers expressed some form of engagement in ACP, but understandings were limited and divergence was expressed about the timing of more extensive conversations about future care. Most patients preferred focusing on the present and suggested their families did not require direction. This placed families in the complex dilemma of protecting their loved ones while compromising their own needs for dialogue. 

In their qualitative study, Fried et al. (2020) [46] found that no advance care plan document was completed by patients or carers. Further, both identified some barriers to ACP:lack of knowledge of the disease trajectory and the potential medical decisions;lack of interest in planning as the patient will not be conscious of decisions;need to ‘stay focused in the present’;belief that family carers would pay attention about issues.

Van Rickstal et al. (2019) [47] reported that family caregivers had limited engagement in ACP. They considered ACP not useful and the reasons suggested were:behavior hindering discussion;adopting a day-to-day attitude;caregivers’ need for self-protection;questions regarding patient’s cognitive capacity for ACP.

In addition, carers reported that ACP should be initiated timely, possibly by a physician.

Van Rickstal et al. (2020) [48] investigated the differences and similarities in ACP perceptions between Belgian and American patients with young-onset dementia and their carers. The similarities regarding the ACP conceptualization were: limited ACP knowledge;little communication about ADs;their recommendation for HPs to timely initiate ACP.

The differences suggested were: EOL decisions encompassed by nation’s laws (euthanasia);more American caregivers drew attention on financial issues than Belgian respondents;in communication about ADs, Belgian caregivers relied on physicians, whereas American caregivers relied on lawyers.

#### 3.2.2. ACP Facilitation and Barriers/Challenges

The studies reported here were conducted in early-stage dementia. Sellars et al. (2019) investigated the views of people with dementia and their carers concerning ACP and EOL care [49]. They included 84 studies and identified 5 themes: circumventing dehumanizing care and treatment (e.g., refusing futile treatments);facing emotionally difficult discussions;navigating existential tensions;lacking confidence in care settings;recognizing personal autonomy.

Participants had trouble in making treatment decisions in the context of ACP and EOL care. 

In their review, Phenwan et al. (2020) identified factors affecting the decisions to initiate ACP amongst people with dementia [50]. Key facilitators were: care settings with supportive guidelines and policies;HPs and carers having a positive relationship with patients;HPs trained on ACP.

Key inhibitors were: lack of knowledge about dementia and ACP;lack of knowledge about the timing to initiate an ACP conversation.

In a mixed methods study, Lee et al. (2020) pilot tested an ACP framework in primary care, reporting participants’ experience with the framework and impacts on clinical practice associated with its use [51]. This framework aimed to enable a shared understanding of personal preferences/goals for future decisions. Results show that people with dementia, surrogatee decision makers, and HPs valued the opportunity for ACP discussions, at the same time HPs were hesitant to use this framework due to lack of time, training, and resources.

Goossens et al. (2020) [52] assessed shared decision-making levels during ACP conversations between HPs and people with dementia living in nursing homes. The shared decision-making levels reported by HPs and patients were high, whereas external raters reported low levels. Only a quarter of patients referred to ACP as the topic of their conversation. These results pointed to the need of HP training on shared decision-making.

Dassel et al. (2019) [53] developed an EOL planning guide which aimed HPs to capture the EOL care values and preferences of people with dementia. Both qualitative and quantitative data supported the utility and feasibility of the guide, which can facilitate the discussion/documentation about EOL care preferences before patients lose decisional capacity. 

##### (1) HP Attitude towards ACP 

All the studies reported here were conducted in advanced dementia.

In their review, Beck et al. (2015) [54] included 14 studies, identifying four themes regarding HP perspectives on ACP: early planning and integration for palliative care;moral and ethical concerns regarding ACP as perceived by HPs;challenges in communicating with patients and carers;need for education and training, particularly in the long-term care setting.

Ethical dilemmas experienced by HPs were also thoroughly reported in the meta-review of systematic reviews by Keijzer-van Laarhoven et al. (2020) [55]. Here, five themes were identified: non-maleficence and beneficence;relationship and courage;responsibility and ownership;respecting dignity.

These themes could be considered either facilitators or barriers for HPs to implement ACP, depending on the context. 

In their qualitative study, Perin et al. (2020) [56] explored physicians’ perceptions regarding ACP and reported their needs and difficulties. Four overarching themes were identified: shaping the healthcare relationship: moving with uncertainty and difficulties;role of family members;ethics vs. pragmatism: making the right decision at the right time;physicians’ needs to improve ACP in daily practice.

Additionally, results showed that physicians had difficulties synchronizing the time of shared ACP with the more rapid development of dementia, and that further HP training on communication on ACP is needed.

In a survey including 261 HPs, Lee et al. (2018) [57] reported moderate levels of knowledge and high levels of interest in ACP. Barriers to ACP were identified as being a lack of time, lack of training, and competing priorities. 

#### 3.2.3. ACP Documentation

Konttila et al. (2020) [58] retrospectively evaluated changes in physician treatment orders, possible burdensome interventions, and symptom prevalence among nursing home patients with advanced dementia who died between 2004–2009 and 2010–2013. The physician orders related to forgoing antibiotics/hydration/artificial nutrition/hospitalization doubled between periods. The physician orders were also completed significantly earlier in 2010–2013 rather than in 2004–2009. The number of physician orders did not affect burdensome interventions or symptom prevalence.

Vandervoort et al. (2014) [11] assessed the effect of written ADs and family physician orders on the quality of dying. They found a strong association between ADs completion and the quality of dying, in particular with lower levels of psychological distress. Nurse-carers communication (instead of nurse-patient) about ACP was negatively associated with the quality of dying.

Pettigrew et al. (2019) [59] investigated factors influencing decision making, preferences regarding ACP and EOL care among people with dementia as reported by their caregivers, and examined differences by race. The 431 participants had a high knowledge on EOL care, as well as high levels of formal/informal ACP discussions. African American participants reported a lower rate of AD completion, lower preference for comfort care, compared to White Americans. 

Moss et al. (2018) [60] evaluated the number of informal/formal EOL care plans among African American older people with dementia. They found that 57% of patients expressed verbal wishes for EOL, while 88% caregivers had at least one document/verbal information about EOL care for their care recipient or at least there was an assigned surrogate.

#### 3.2.4. ACP Interventions 

All the studies reported here are pre-post intervention studies.

##### (1) Interventions for HPs

Using real case scenario, Katwa et al. (2020) [61] evaluated a simulation training for care home workers. This one-day training was well-received by participants, showing benefits of shared learning and a better understanding of multi-disciplinary working with other HPs.

In a quasi-experimental study, Ampe et al. (2017) [62] pilot tested the ‘weDECide–Discussing End-of-life Choices’ intervention (vs. usual care) on nursing home staff of a dementia care unit. After the intervention, the ACP policy was significantly more adherent with best practice, whereas the policy in the control group was not. ACP was not discussed more frequently, nor were patients and carers involved more in conversations. ACP facilitators were: support by management staff, and involvement of the whole organization. Barriers to ACP implementation were staff’s limited responsibilities.

Cotter et al. (2019) [63] assessed the effectiveness of a one-hour educational intervention on physician (N = 16) knowledge, attitudes, and skills, and prevalence of ACP documentation in the electronic medical record. After the intervention, there was an improvement in the ability to discuss ACP, belief that ACP improves outcomes in dementia, knowledge about ACP Medicare billing codes. AD and medical orders for life-sustaining treatment rates increased in both people with and without dementia.

##### (2) Intervention for Patients and Carers

Huang et al. (2020) [64] investigated the efficacy of a nurse-led family-centered ACP information intervention in 40 people with dementia-caregivers dyads in Taiwan. After the intervention, the dyads were significantly more knowledgeable about ACP and dementia treatment, showed a significant reduction in decisional conflict on mechanical ventilation, cardiopulmonary resuscitation, and tracheostomy. There were no changes in attitude toward ACP for people with dementia, whereas there was a reduction in negative attitude for caregivers.

##### (3) Randomized Controlled Trials

We included ten RCTs with a total of 14,794 participants [32,33,34,35,36,37,38,39,40,41]. 

Setting and participants—The majority of studies were conducted in nursing homes [32,35,36,37,40,41] and involved patients with severe dementia and/or carers. One study only involved HPs of patients with early dementia [36]. 

The remaining studies were conducted in community/outpatient settings [33,34,38,39] involving patients and carers of people with severe dementia [38], patients and carers of people with moderate to severe dementia [39], carers of patients with moderate to severe dementia [34], and patients, carers and HPs of patients with any type of dementia [33], One RCT was conducted in an acute hospital [37].

Description of interventions - Intervention modalities included training programs for HPs (20%), videos (30%), educational programs (20%), facilitated discussions (20%), and written-only materials (10%) for patients and/or carers. We did not find any study performed using web-based platforms.

Hanson et al. (2017) assessed the efficacy of a 18-min *video* decision aid vs. an informational video on interaction with someone with dementia [40,65] in 302 people with advanced dementia and their carers from 22 nursing homes. Mitchell et al. (2018) assessed the efficacy of a 12-min educational *video* vs. usual ACP practice [37] in 402 patients with dementia and their carers from 64 nursing homes. Mitchell et al. (2020) assessed the efficacy of five 6- to 10-min *videos* [32] vs. usual ACP practice in patients and/or carers of patients with severe dementia from 360 nursing homes. 

Development of the videos was not reported in two studies [32,40], whereas in the other the intervention was a refined version of the EVINCE, whose details were reported elsewhere [37]. Contents of the videos dealt with goals of care in two studies [32,40]. In the other the intervention consisted of a video on ACP for carers and a form delivered to the HPs indicating the carer’s preferred level of care after watching the video [37]. Providers of the intervention were the nursing home care team, a researcher, and designated champions (mostly social workers).

The two studies providing *training programs for HPs* (vs. no training) consisted of two workshops (3 h each involving 38 family physicians) [33], and two workshops (4 h each) with homework assignments, involving 311 HPs [36].

One intervention also provided supporting materials to all participants in order to facilitate the assimilation of the model [36]. In Tilburgs et al. (2020) [33], the trainer was a family physician and researcher, and in Goossens et al. (2020) [36] the trainers were one psychologist and one nurse. 

Both interventions were based on the shared decision-making model. In the first, two workshops included the presentation of such model, real-life case scenario, ACP conversation practice, and its documentation in medical files [33]. In the other study, Goossens et al. (2020) [36] provided information on goals of prolonging life, dementia, comfort care, treatments consistent with each goal, and how to prioritize goals.

Two studies were on *educational programs.* Bonner et al. (2020) [34] assessed the efficacy of a 4-week class (1-h) plus a booster session (vs. attention control) in 358 carers of people with dementia. Song et al. (2019) assessed the efficacy of a psychoeducational intervention (i.e., SPIRIT) in person vs. SPIRIT remote in 23 patients with dementia-carers dyads [39]. The SPIRIT was provided by two HPs, and one social worker, respectively [39].

In Bonner et al. (2020) [34], the intervention was developed using the Kolb’s Experiential learning theory, and the Theory of Reasoned Action was used to select the outcome measures. In Song et al. (2019) [39], the intervention was developed using the representational approach by Donovan et al. (2007) [66]. 

The contents of the program delivered by Bonner et al. (2020) dealt with the disease trajectory of dementia, as well as risk/benefits of cardiopulmonary resuscitation, mechanical ventilation, and tube feeding [34]. Song et al. (2019) [39] delivered the SPIRIT intervention guide which has been developed and reported in detail elsewhere. This guide aimed to thoroughly assess patient issues about their own illness, and assist them in evaluating their values related to EOL. In addition, the guide allows the surrogate to better understand the patient’s illness experiences/values and to be prepared for the responsibility that can arise during EOL decision making. A ‘goals of care’ tool is completed right at the end of the session to indicate the patient’s preferences [39].

Two studies provided *facilitated discussions* to patients and/or carers of patients with severe dementia. Brazil et al. (2018) assessed the efficacy of two family meetings (1-h each) [35] plus usual care (vs. usual care alone) in 695 patients from 24 nursing homes. Sampson et al. (2011) assessed the efficacy of up to four discussions (vs. usual care) in 32 patient-carer dyads [38]. Providers were two senior nurses in both studies. One intervention was developed by means of clinical guidelines for leading family meetings in the context of palliative care [67]. The other intervention used the Medical Research Council framework for developing and testing complex health interventions. In Brazil et al. (2018) [35] after reviewing the booklet contents, the ACP facilitator assisted the family carers to think over patient values, goals, and EOL care, so as to facilitate best interest decision-making. Then, the carers reviewed a provisional advance care plan based on the former discussion, to address any issues, and sign the standardized advance care plan document. Lastly, the advance care plan was placed in the patient medical record, and a copy sent to the patient’s family physician.

In the study by Sampson et al. (2011) [38] the facilitators extensively assessed patients’ knowledge and severity of dementia, their palliative care needs, and whether they had completed any ADs/statements. In the subsequent consultations, the facilitators provided information on dementia, its prognosis, and palliative care. 

Saevareid et al. (2019) [41] assessed the efficacy of an *ACP guideline* implementation in 154 patients (8 nursing homes). The guideline emphasized voluntary participation and recommended inclusion of patients with cognitive impairment. It also included a 2-day training seminar for the project team (who would then train the other staff at the ward), and a pocket card for spontaneous conversation and template on how to document ACP.

*Outcomes—*Overall, there were 43 outcomes (14 primary and 29 secondary) across the 10 studies, including 7/43 process, 14/43 action, 8/43 quality of care, 7/43 health status, and 7/43 healthcare utilization outcomes.

*Interventions with positive primary outcomes—*Of the primary outcomes, three were on process, four on action, four on quality of care, and one was on healthcare utilization. One of 15 outcomes for videos was positive, as were 2/6 for training programs for HPs; 2/5 for educational programs, 2/15 for facilitated discussion, and 1/5 for written only materials.

Five of seven process outcomes were positive: knowledge of (and self-efficacy for) mechanical ventilation, cardiopulmonary resuscitation, and tube feeding treatment decisions [34], as well as the level of shared decision-making during formal ACP conversations, perceived importance, and competence [36]. 

Among action outcomes, 8/14 were positive. Specifically, all dealt with the documentation of value and preferences: proportion of people with dementia who had at least one ACP conversation documented in their medical file [33]; number of carers making an ACP [38]; patient participation in EOL treatment conversations; elicitation of the patient’s own expressions of future preferences, hopes and worries and the patient’s competency to consent [41]; medical order for scope of treatment completion [40].

Three of eight quality of care outcomes were positive. One dealt with the communication (i.e., assessed with the quality of communication questionnaire [40]). The other two were satisfaction with care and decision-making outcomes [35].

One of the seven health status outcomes was positive [40].

Among seven healthcare utilization outcomes, only two were positive: hospital transfers were decreased [40], number of nonmedical (i.e., social contacts, activities), and medical (i.e., hospital admission, resuscitation) preferences discussed were increased [33].

### 3.3. Amyotrophic Lateral Sclerosis/Motor Neuron Disease

Benditt et al. (2001) [68] described a disease-specific ACP document including mechanical ventilation and nutrition. This document provides a catalyst for discussion among patients and carers and serves as a framework for future decisions (an AD form is also included). Hossler et al. (2011) [69] assessed in a pre-post pilot study the feasibility of ‘Making Your Wishes Known’, an interactive, computer-based ACP decision aid. After the intervention, 16/17 patients completed the computer-based ADs. No burden was reported by patients; satisfaction with, and perceived accuracy of the intervention were high. The information provided was considered appropriate. The intervention prompted many participants to discuss ACP with family members and to share their ADs with their physician. 

Using an online version of the same ‘Making Your Wishes Known’ decision aid reported above, Levi et al. (2017) [70] assessed its impact on patient-clinician communication regarding EOL wishes. After the intervention, there was a statistically significant improvement in the concordance between patient wishes and HP decisions, and the HPs were more confident in that their decisions accurately represented each patient’s wishes. Further, patients reported high satisfaction and low decisional conflict with decisions about EOL care, and high satisfaction with the decision aid. Further, patient knowledge about ACP increased post-intervention.

Murray et al. (2016) [71] systematically assessed the content, prevalence, patient/carer benefits, HP awareness/support, and outcomes associated with ACP. ACP prevalence varied considerably across the 16 included studies, due to different follow up as well as geographical factors. Among ACP predictors, disease progression was considered the strongest catalyst for AD completion. EOL decisions were influenced by patient healthcare goals and clinical circumstances.

Murray et al. (2016) [72] qualitatively evaluated the caregiver perceptions on the impact and acceptability of ACP, by means of a letter format, for motor neuron disease patients and their carers. Four themes emerged: empowerment;readiness for death;clarifying choices;decisions.

Carers deemed the letter of future care as beneficial, by improving autonomy both for patients and themselves. Appropriate timing for ACP initiation was considered to strictly depend on patient characteristics.

Preston et al. (2012) [73] qualitatively investigated the experience of bereaved relatives on the Preferred Priorities for Care document. Despite completion of the document being emotionally difficult, positively affected the EOL care for all the participants. Additionally, its completion was associated with a sense of relief, and helped the patient to maintain a sense of control. Timing was identified as a key challenge of the ACP process. Notably, there was lack of awareness of the document in clinical practice.

Seeber et al. (2019) [74] interviewed 21 Dutch ALS outpatients on the current practice of ACP initiation soon after diagnosis communication. In this approach, the neurologist usually gives a general outlook of the disease and then the patient is introduced to a multidisciplinary team. Iterative and tailor-made discussions on future treatments are performed during follow up by means of regular appointments. The study shows that this practice is well-accepted by ALS patients. Based on these empirical findings, authors formulated and discussed some recommendations about the integration of ACP in the care of patients with other chronic neurological diseases. The first recommendation is that ACP should be initiated from diagnosis onwards; second, following up patients could facilitate the ACP maintenance; third, as ACP is a professional skill, HPs should be trained in dealing these conversations.

### 3.4. Brain Tumors

By analyzing qualitatively the Brain Tumor Social Media tweet chat about ACP, Cutshall et al. (2020) [75] derived three themes from the key stakeholders’ perspective:barriers preventing EOL discussions;need to assure that patient perspective was considered;right time for ACP.

Twitter (as other social media) was considered an opportunity for all the stakeholders to better understand each other’s perspectives related to ACP.

Fritz et al. (2020) [76] conducted a qualitative study to develop an ACP program specifically for glioblastoma patients. They explored topics and practical issues which were relevant for patients and their proxies, together with barriers and facilitators to participate in an ACP program. While there was a consensus between the participants on the program contents, debate remained about the appropriate timing to introduce such a program. A barrier to participate was that the program was considered too challenging, whereas access to information was deemed a facilitator.

Llewellyn et al. (2018) [77] involved key HPs working in neuro oncology in order to investigate their experiences and assumptions on ACP. Few HPs had completed a formal ACP document. Eight key factors were identified: lacking time and patient contact;emotive conversations;windows of opportunity;professional remit;personality and rapport;professional identities and perceived expectations;constitutive practices of ACP;shared responsibility.

These factors contributed to three main conditions for avoidance: difficulties with ACP practice; ambiguities in ACP definition/scope/practice; and availability and presence of others. With their interaction, these three factors produce a ‘culture of shared avoidance’.

Pollom et al. (2018) [78] performed a retrospective chart review of newly-diagnosed glioblastoma patients, treated with radiation. Half of the patients had an ACP documented at the time of the last follow-up, with one third of patients having ACP documented within six months since their diagnosis. Of the 44 deceased patients, 24 had ACP documented before death. Only 11 patients had received ACP education.

In a systematic review including 19 studies (486 patients with primary brain tumors), Song et al. (2016) [79] found that ACP conversations were rarely used at the EOL. AD completion and place of death rates differed between studies. ACP was significantly associated with lower intensive care unit utilization and hospital readmission rates. A RCT assessed the efficacy of a ‘goals of care’ video decision support tool after a verbal narrative (vs. verbal narrative alone) in improving ACP. The video proved effective in gaining confidence in decision-making and promoting comfort care. Nevertheless, the effect of the intervention on care at the EOL and QOL were unclear [80].

### 3.5. Parkinson’s Disease 

Tuck et al. (2015) [81] investigated the preferences of Parkinson’s disease patients about the initiation and timing of conversations related to prognosis, and ACP/EOL care options. Patients preferred that their carers be involved early in EOL conversations. Further, half desired to talk about ADs early in the disease course whereas many (25%) desired to postpone discussions on life expectancy and practical aspects of EOL care until their condition worsened. A small percentage (12%) would have discussed EOL issues at the time of diagnosis. The majority of participants (69%) signed an ACP document. 

With the aim to inform a patient and carer-centered framework for Parkinson’s disease clinical care and research, Lum et al. (2019) [10] explored the patients’ and carers’ perspectives on ACP. Four themes emerged: personal definitions of ACP varied in Parkinson’s disease (e.g., ACP as part of routine care before diagnosis of Parkinson’s disease);barriers for ACP engagement were related to the health care system (e.g., patients’ lack of trust that ACP preferences would be honored by HPs), patient cognitive decline, and difficult relationship within the family;carers role in ACP (e.g., assuming the role of surrogate decision maker);positive influence of a palliative care approach for ACP initiation.

### 3.6. Duchenne Muscular Dystrophy

Abbott et al. (2017) [82] in their qualitative study investigated the views and preferences of 15 Duchenne muscular dystrophy patients about EOL planning. Participants did not remember any significant conversations with HPs about EOL, and hypothesized that HPs were unwilling to discuss this issue. Patients also desired to have more information about causes of death and EOL management, and practical and emotional support for funerals, place of death, and to discuss these issues with their families. 

### 3.7. Multiple Sclerosis

Cottrell et al. (2020) [9] in their realist review, aimed to identify contextual factors influencing core mechanisms, and to contribute to the theoretical understanding of ACP in MS. By using the Integrated Behaviour Model, they identified as core mechanisms the patients’ acceptance of previous experiences, their situation, confidence, desire for autonomy, and fear. In addition, self-acceptance emerged as key allowing patients to consider ACP as pertinent for them. The MS care context hindered triggering of the above-mentioned mechanisms. Absence of HP communication skills was found to be a barrier to ACP implementation.

### 3.8. Mixed Populations

Walter et al. (2019) [20] involved 125 neurologists in an online survey aimed to assess content and timing of discussions on treatment restrictions (initiate/withhold/withdraw treatment) in people with high grade gliomas, MS, and Parkinson’s disease. Findings show that in people with Parkinson’s disease and MS discussions on treatment restrictions are initiated later compared to patients with high grade gliomas. The trigger for EOL care was usually the patient physical and cognitive decline. The majority of participants reported the need to be trained in EOL discussions.

In a retrospective chart review, Cheung et al. (2017) [83] investigated the EOL care communication, preferences and documentation among patients with end-stage neurological disorders (stroke, myopathies, motor neuron disease, MS, parkinsonism). Results showed that most patients would decide for EOL issues during their first consultations. For patients who signed ADs, more frequently cited EOL interventions were cardiopulmonary resuscitation, mechanical ventilation, artificial nutrition, and hydration. For patients who had ACP only, the most common diagnosis was stroke; artificial nutrition, hydration, and place of death were the most reported EOL issues.

## 4. Discussion

ACP has been defined as a process consisting of many behaviors, such as indicating a surrogate decision maker, defining preferences and values for medical and EOL care, and communicating those wishes to others [27,84].

In this scoping review we mapped the body of evidence on ACP (interventions) in neurodegenerative disorders. After screening 9367 references, we included 53 studies. The majority of studies (and all the RCTs) were conducted in dementia, followed by ALS/motor neuron disease (13%), and brain tumors (9%).

Overall, in absolute numbers, US is the country where most studies have been conducted, followed by—among European countries, UK, the Netherlands, and Belgium. These findings mirror the adoption of ACP initiatives within healthcare systems, including payer reimbursement programs and quality metric initiatives in the US, besides the implementation of the Patient Self-Determination Act in 1990 [85]; and for the other countries the adoption of legal ADs (e.g., Mental Capacity Act in England) in the last three decades [86,87,88].

More than 60% of the included studies referred to ACP as a process and EOL planning discussions. We based our inclusion criteria on the ACP definition by Rietjens et al. (2017) [5] who tried to overcome earlier definitions, by shifting from eliciting treatment instructions to be used when an individual’s decisional capacity has been lost towards communication about goals and preferences for future medical care across the spectrum of ages and illnesses [5].

In the context of neurodegenerative disorders, ACP is influenced by the difficulties to predict the time and features of the disease evolution. As such, the discussion and communication of the patient’s decision should be possibly planned early in the disease course, as recommended by Oliver et al. (2016) in the EAN consensus on palliative care in chronic and progressive neurological diseases [89,90]. In addition, there is a lack of reliable end stage indicators to help predict the last months of life in these disorders [89,90].

Our findings show that the appropriate timing to initiate an ACP conversation may differ between the different neurodegenerative disorders, and depend on the specific disease trajectory, and prognosis. In people with Parkinson’s disease and MS, ACP conversations start later in comparison to brain tumors, and ALS/motor neuron disease [74]. Further, together with physical problems, a common barrier as well as a trigger to initiate ACP across these neurodegenerative disorders is the patient’s cognitive capacity. This is particularly the case in dementia [91], but also in the other disorders, such as brain tumors where a recent study aimed to determine the medical decision-making capacity in these patients. Lack of capacity at large may limit the patient ability to give free and informed consent to medical treatments or research [92], and this can be also applied to ACP. Furthermore, a common facilitator which has been mainly identified in dementia [42, 53] and could be easily extended to the other disorders is the training of HPs in dealing ACP conversations. HPs often fail to initiate these conversations due to their reluctance to discuss disease progression and EOL issues; fear of distressing patients and making them to lose hope; and difficulty managing own emotions [9].

Due to study heterogeneity and different outcome measures used, it was not possible to perform meta-analysis of the RCTs we found, all of which concerned dementia. However, we thoroughly investigated the study characteristics extracting and reporting data on the intervention components and relevant outcomes.

Recently, there has been much debate on which outcomes may successfully capture the ACP process, and some experts have developed an ACP Outcomes Framework [27]. Based on this framework, process, quality of care and action were the most used outcomes within the included RCTs in dementia. Specifically, 71% of the process and 57% of the action outcomes were positive. These proportions are consistent with those reported by McMahan et al. (2020) [26] in their scoping review that included populations not restricted to dementia. Another important consideration is that measuring ACP outcomes could be difficult. In fact, even if goal concordance was considered to be the gold standard, it is generally hard to measure it, as for example preferences can change over time, thus reliance on ACP documentation can be in-accurate [26].

This review has some strengths in that, to increase depth and decrease the risk of missing important information on ACP in neurodegenerative disorders, we decided to include both (systematic) reviews and primary studies. Most studies reported research on dementia, whereas a few on other neurodegenerative disorders. The body of evidence deriving from ALS shows a few examples on the use of an ACP intervention [69,70], and also reports interesting findings from a Dutch outpatient clinic [74], which warrant further confirmation. Finally, some efforts are being made to develop an ACP intervention for brain tumors [75] and further testing of efficacy is part of an ongoing project.

We based our inclusion on the most recent ACP definition and categorized outcomes included in the RCTs using the above-mentioned ACP Outcomes Framework [27]. 

Despite these strengths, this study has some limitations. As this was a scoping review, our search strategy was not comprehensive, as it was restricted to PubMed, Cochrane Database of Systematic Reviews, Cochrane Central Register of Controlled Trials, and PROSPERO, and did not include other biomedical research databases. Further, the results were limited to the English language and originated from higher-income countries having supportive ACP laws and policies, and mostly from patients with a jewish-christian cultural background. Thus, the transferability of our findings beyond these populations is unclear.

Although not envisaged in the scoping review methods, we assessed the quality of studies using different tools (e.g., CASP tools), as different designs were included. However, we acknowledge that using a unique tool would be preferable in order to make appropriate comparisons across different study designs.

## 5. Conclusions

To conclude, more research is needed investigating barriers and facilitators of ACP in neurodegenerative disorders. An early initiation of ACP is proposed, also in conditions with a long disease course, such as MS and PD. Training programs for HPs (chiefly physicians and nurses) are key to improve their competences and self-confidence in initiating these conversations in response to a clinical worsening or to a patients’ cue. 

Evidence on the efficacy of ACP interventions is lacking in almost all the chronic neurodegenerative disorders. On this regard, it is worth mentioning that the European Association of Palliative Care recently appointed a task force to further investigate ACP in dementia and to develop recommendations for practice, policy and research [93].

Considering the advances of technologies, future studies should assess the feasibility of ACP programs using new formats [94], and adapting such interventions to the local legal and cultural circumstances and to the disease stage. 

Outcomes should be further investigated by considering the varying stage of change for each discrete ACP behavior [95]. Moreover, outcome measures validated in the different populations (i.e., diseases and cultures) and standardized across studies are needed.

## Figures and Tables

**Figure 1 ijerph-19-00803-f001:**
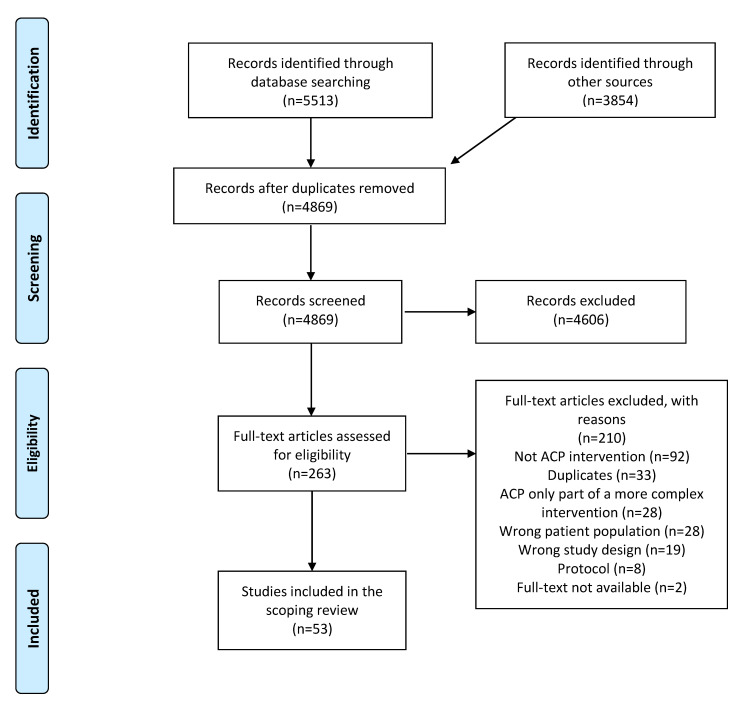
Study flow diagram. ACP is advance care planning.

**Figure 2 ijerph-19-00803-f002:**
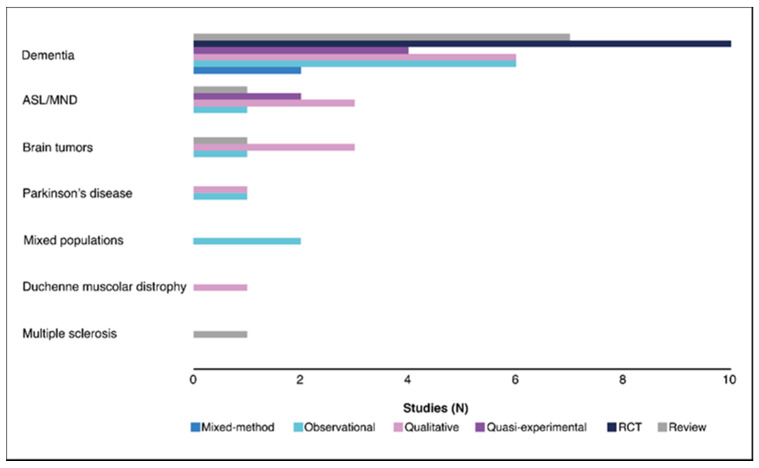
Number of studies by disease and by study design. ALS is amyotrophic lateral sclerosis; MND is motor neuron disease; RCT is randomized controlled trial.

## Data Availability

Data sharing not applicable. No new data were created or analyzed in this study.

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
