# Peer review of "Advance Care Planning in Neurodegenerative Disorders: A Scoping Review"

_ijerph, 2022, doi:10.3390/ijerph19020803_

Round 1

Reviewer 1 Report

Excellent work, massive literature search and extensive review of selected trials. I would like to see with each trial discussed how many participants were included.

One weakness is briefly discussed: cultural context. It might be appropriate to stress in the introduction how important cultural context is for ACP and in the discussion that the results discussed are from higher income countries, but also from patients with a jewish-christian cultural background. Was there mention of any other patients (i.e. european muslims, afro-american or american asian background) in the reviewed trials?

Author Response

Excellent work, massive literature search and extensive review of selected trials. I would like to see with each trial discussed how many participants were included.

We thank the reviewer for his/her appreciation of our work. We have now reported in the main text the number of participants for each trial.

One weakness is briefly discussed: cultural context. It might be appropriate to stress in the introduction how important cultural context is for ACP and in the discussion that the results discussed are from higher income countries, but also from patients with a jewish-christian cultural background. Was there mention of any other patients (i.e. european muslims, afro-american or american asian background) in the reviewed trials?

We thank the reviewer for this comment. We have now stressed in the introduction the importance of cultural context in the ACP process, and added in the discussion that our findings derive from higher income countries, and from patients with a jewish-christian cultural background.

Reviewer 2 Report

This article contains a lot of relevant and interesting information about the topic area, but it sometimes felt a bit difficult to read. I hope the comments below are constructive and will help to make it a bit more accessible to the reader.

General questions and points to address or consider

If you’re looking at the ‘current’ stage of research, why was your search not focused on the most recent studies? I know 87% of the ones you were included were published in the last 5 years, but is anything older than 5 years still ‘current’? it’s not a criticism, more of a query that arose when I was reading the article. It might be worth providing some explanation/justification for not setting a stricter date range.

Figure 1 – it’s slightly confusing using ‘duplicates’ in the eligibility stage as the diagram implies you removed duplicates earlier on. I assume it means articles reporting on the same study, rather than the same article, but it could be worth clarifying this. It’s also a bit of a surprise as it’s not one of the exclusion reasons given in lines 169-171 (where you could have explained it).

The article is tricky to follow at times due to some fairly long and convoluted sentences, often with many clauses. It might be worth trying to split some of the longer sentences into two shorter ones. Also, rather than listing themes separated by ; or , it might be worth considering bullet lists in some cases to make them clearer to read. For example, the six themes in lines 243-247, especially as they are numbered. It can also be confusing to have a mix of ; and , in the same list such as lines 573-575 as you’re not sure where a theme ends. The frequent use of putting clauses or examples in () can make sentences harder to read. See lines 122-142 for multiple examples. I understand that it’s necessary at times, but it can be annoying or distracting for the reader as it makes it easier to lose the meaning of the sentence. It doesn’t help when you have an example for each point, and use i.e. twice within the same point, see lines 618-624.

Lines 386-388 – I’m confused by the sentence ‘After the intervention…conversations.’ I don’t understand how the ACP policy can be significantly more adherent with best practice, but not result in more discussions or involvement in conversations. Am I missing something? Were these aspects not part of the policy? I think something should be done to clarify this bit, otherwise it could leave the reader with more questions than answers.

Lines 404-405 – Similar to the above, I’m not sure how there can be no change in attitude but a reduction in negative attitude for caregivers. If it’s attitudes towards different things or by different people, that needs to be made clearer.

I find it odd than in places you give results as both ‘one out of 15’ and ‘7%’ but in other places you don’t. For example lines 491-497 compared to lines 500-516. It seems inconsistent. I personally prefer without the % as it makes it easier to follow, but don’t particularly mind which you do, as long as you are consistent. Think about why you are presenting results in a particular way. Does the % add anything to what you’re saying?

Lines 594-598 are a bit odd, as you talk about ‘newly-diagnosed’ at the start, but we quickly move on to ‘deceased patients’. I can’t quite work out who the study is actually focusing on.

Lines 627-635 – was there anything mentioned about the fact that all participants were men? For example, acknowledging that findings might not be generalisable as not have female perspective?

I’m not sure about your wording ‘except dementia’. Just because a lot of work has been done in this area doesn’t mean it should be excluded in the future, especially as many outcomes were not positive, e.g. only 3 of 8 quality of care outcomes, 1 of 7 health status outcomes, 2 of 7 healthcare utilization outcomes. Maybe a slightly softer wording to be used to emphasise a priority or focus on non-dementia conditions rather than ruling it out altogether.

Minor points

Line 55 – I think it should be either ‘with patient needs’ or ‘with the patient’s needs’ rather than ‘with the patient needs’

Line 139 – why has ‘Care’ got a capital C?

Line 144 – ‘For the (systematic) reviews was assessed’ doesn’t make sense. Should it be something like ‘The (systematic) reviews were assessed’?

Where is Figure S2? There was a risk of bias summary but that was labelled Figure S1.

Line 315 – as far as I can tell you haven’t said what the abbreviations SDM and NH stand for

Lines 339-343 – the mix of : and ; in one sentence makes it difficult to follow and work out what the four overarching themes actually are. For example, is ‘moving with uncertainty and difficulties’ linked to ‘shaping the healthcare relationship in the dementia setting’ or is it separate?

Line 349 – for me, ‘Barriers to ACP were identified in that lack of time…’ doesn’t make sense. ‘Barriers to ACP were identified as being a lack of time…’ might be better

Line 371 – it should be American not Americans

Line 375 – what is a ‘verbal document’?

Line 430 – what do you mean by ‘providers’? Providers of what? It’s not clear to me.

Lines 433-434 – I think it should either be ‘two workshops (3 hours each) [32], and two workshops ( 4 hours each)’ or two 3-hour workshops [32], and two 4-hour workshops’.

Lines 436-437 – you should be using ) not ] after the dates

Line 456 – I think it should be detail not details

Line 469 – I think the date should be in ()

Line 496 – Sorry, but 2/6 is rounded down to 33% not up to 34%

Line 554 – I think it should be ‘being’ not ‘was’

Line 575 – I’m not sure what you mean by ‘(as others social media)’. As other social media or as with other social media?

Line 582 – should it be ‘such a program’ or ‘such programs’ rather than ‘such program’?

Lines 613-614 – ‘as long as their condition worsened’ doesn’t seem to fit in the context. Do you mean ‘until their condition worsened’?

Line 615 – The majority, not majority

Line 698 – You don’t need ‘such as’ and ‘e.g.’. ‘such as brain tumors’ would be fine

Author Response

General questions and points to address or consider

If you’re looking at the ‘current’ stage of research, why was your search not focused on the most recent studies? I know 87% of the ones you were included were published in the last 5 years, but is anything older than 5 years still ‘current’? it’s not a criticism, more of a query that arose when I was reading the article. It might be worth providing some explanation/justification for not setting a stricter date range.

We thank very much the reviewer for this comment.  In this work, we aimed to assess the stage of the research in this field in general, so as to promote future research and inform clinical practice. To better align and to be consistent with the time frame of the bibliographic search, we removed the term ‘current’ from the main text, as it was a typo.

Figure 1 – it’s slightly confusing using ‘duplicates’ in the eligibility stage as the diagram implies you removed duplicates earlier on. I assume it means articles reporting on the same study, rather than the same article, but it could be worth clarifying this. It’s also a bit of a surprise as it’s not one of the exclusion reasons given in lines 169-171 (where you could have explained it).

We thank the reviewer for this comment. First, three duplicates of the identified records were removed as they were report for the same article. Second, in the selection of the full-texts, 33 duplicates were removed from the final list, as reported in the Figure 1. To be consistent with this Figure, we added in the results also this detail, as follows: ‘[…] Of the 263 full-text articles retained for further screening, 210 were excluded because they were not focused on ACP, or ACP was only part of a more complex intervention, had wrong patient population, study design, were protocols, duplicates, or with no full-text available’.

The article is tricky to follow at times due to some fairly long and convoluted sentences, often with many clauses. It might be worth trying to split some of the longer sentences into two shorter ones.

We thank the reviewer for this comment. We have now tried to improve the readability of the manuscript, adding bullet points when appropriate, and revising some difficult and convoluted sentences along the text.

Also, rather than listing themes separated by ; or , it might be worth considering bullet lists in some cases to make them clearer to read. For example, the six themes in lines 243-247, especially as they are numbered. It can also be confusing to have a mix of ; and , in the same list such as lines 573-575 as you’re not sure where a theme ends. The frequent use of putting clauses or examples in () can make sentences harder to read. See lines 122-142 for multiple examples. I understand that it’s necessary at times, but it can be annoying or distracting for the reader as it makes it easier to lose the meaning of the sentence. It doesn’t help when you have an example for each point, and use i.e. twice within the same point, see lines 618-624.

To improve readability, we have now added bullet points when citing themes along the results.

Lines 386-388 – I’m confused by the sentence ‘After the intervention…conversations.’ I don’t understand how the ACP policy can be significantly more adherent with best practice, but not result in more discussions or involvement in conversations. Am I missing something? Were these aspects not part of the policy? I think something should be done to clarify this bit, otherwise it could leave the reader with more questions than answers.

We have now slightly changed the text, in order to improve the reporting of this study, as follows: ‘In a quasi-experimental study, Ampe et al. (2017) [62] pilot tested the ‘weDECide–Discussing End-of-life Choices’ intervention (vs. usual care) on nursing home staff of a dementia care unit. After the intervention, the ACP policy was significantly more adherent with best practice, whereas the policy in the control group was not. ACP was not discussed more frequently, nor were patients and carers involved more in conversations.’

In addition, study authors reported that ‘In practice, ACP appeared not to be discussed more frequently or on a higher level than before ‘we DECide’, and residents and families were not involved in conversations to a greater extent. Although we expected this for the control group, we did not for the intervention group. We hypothesized that the policy as well as the actual practice would benefit from ‘we DECide’, because both management and clinical staff were involved, and because the intervention was a very practically oriented training.’

Lines 404-405 – Similar to the above, I’m not sure how there can be no change in attitude but a reduction in negative attitude for caregivers. If it’s attitudes towards different things or by different people, that needs to be made clearer.

We have now slightly changed the text, as follows: ‘There were no changes in attitude toward ACP for people with dementia, whereas there was a reduction in negative attitude for caregivers’.

I find it odd than in places you give results as both ‘one out of 15’ and ‘7%’ but in other places you don’t. For example lines 491-497 compared to lines 500-516. It seems inconsistent. I personally prefer without the % as it makes it easier to follow, but don’t particularly mind which you do, as long as you are consistent. Think about why you are presenting results in a particular way. Does the % add anything to what you’re saying?

We have now slightly changed the text accordingly, removing the percentages along the suggested lines.

Lines 594-598 are a bit odd, as you talk about ‘newly-diagnosed’ at the start, but we quickly move on to ‘deceased patients’. I can’t quite work out who the study is actually focusing on.

This study is a retrospective chart review in which were included. patients with newly-diagnosed glioblastoma.

Lines 627-635 – was there anything mentioned about the fact that all participants were men? For example, acknowledging that findings might not be generalisable as not have female perspective?

As the Duchenne muscular dystrophy mainly affects boys, we believe that no change is needed in the text.

I’m not sure about your wording ‘except dementia’. Just because a lot of work has been done in this area doesn’t mean it should be excluded in the future, especially as many outcomes were not positive, e.g. only 3 of 8 quality of care outcomes, 1 of 7 health status outcomes, 2 of 7 healthcare utilization outcomes. Maybe a slightly softer wording to be used to emphasise a priority or focus on non-dementia conditions rather than ruling it out altogether.

We thank the reviewer for this comment. Based on his/her suggestion and on the comment of reviewer #4, we have now slightly changed the text in the Conclusions, as follows: ‘To conclude, more research is needed investigating barriers and facilitators of ACP in neurodegenerative disorders. An early initiation of ACP is proposed, also in conditions with a long disease course, such as MS and PD. Training programs for HPs (chiefly physicians and nurses) are key to improve their competences and self-confidence in initiating these conversations in response to a clinical worsening or to a patients’ cue.

Evidence on the efficacy of ACP interventions is lacking in almost all the chronic neurodegenerative disorders. […]’

Minor points

Line 55 – I think it should be either ‘with patient needs’ or ‘with the patient’s needs’ rather than ‘with the patient needs’.

We have now corrected this typo.

Line 139 – why has ‘Care’ got a capital C?

We have now corrected this typo.

Line 144 – ‘For the (systematic) reviews was assessed’ doesn’t make sense. Should it be something like ‘The (systematic) reviews were assessed’?

We have now corrected this typo.

Where is Figure S2? There was a risk of bias summary but that was labelled Figure S1.

This was a typo in the original version of the manuscript. In the revised version, the Figure 2 has been moved to the Supplementary Materials (see comment of the reviewer 3), being Figure S1. The original Figure S1 has been renamed to Figure S2.

Line 315 – as far as I can tell you haven’t said what the abbreviations SDM and NH stand for

To help the reader a bit, we have removed such acronyms along the text.

Lines 339-343 – the mix of : and ; in one sentence makes it difficult to follow and work out what the four overarching themes actually are. For example, is ‘moving with uncertainty and difficulties’ linked to ‘shaping the healthcare relationship in the dementia setting’ or is it separate?

We have now slightly changed the text, adding bullet points, so as to better clarify which are the identified themes.

Line 349 – for me, ‘Barriers to ACP were identified in that lack of time…’ doesn’t make sense. ‘Barriers to ACP were identified as being a lack of time…’ might be better
We have now corrected this typo.

Line 371 – it should be American not Americans

We have now corrected this typo.

Line 375 – what is a ‘verbal document’?

We have now corrected this typo, removing ‘verbal’.

Line 430 – what do you mean by ‘providers’? Providers of what? It’s not clear to me.

We have now slightly changed the text, accordingly, as follows: ‘Providers of the intervention were the nursing home care team, a researcher, and designated champions (mostly social workers)’.

Lines 433-434 – I think it should either be ‘two workshops (3 hours each) [32], and two workshops ( 4 hours each)’ or two 3-hour workshops [32], and two 4-hour workshops’.

We have now corrected this typo, as follows: ‘The two studies providing training programs for HPs (vs. no training) consisted of two workshops (3- hours each involving 38 family physicians) [323], and two workshops (4- hours each) with homework assignments, involving 311 HPs [356].’

Lines 436-437 – you should be using ) not ] after the dates

We have now corrected this typo.

Line 456 – I think it should be detail not details

We have now corrected this typo.

Line 469 – I think the date should be in ()

We have now corrected this typo.

Line 496 – Sorry, but 2/6 is rounded down to 33% not up to 34%

We have now corrected this typo.

Line 554 – I think it should be ‘being’ not ‘was’

We have now corrected this typo.

Line 575 – I’m not sure what you mean by ‘(as others social media)’. As other social media or as with other social media?

We have now corrected this typo, as follows: ‘Twitter (as other social media) was considered […]’.

Line 582 – should it be ‘such a program’ or ‘such programs’ rather than ‘such program’?

We have now corrected this typo, as follows: ‘[…] timing to introduce such a program.’

Lines 613-614 – ‘as long as their condition worsened’ doesn’t seem to fit in the context. Do you mean ‘until their condition worsened’?

We thank the reviewer for this suggestion. We have now changed the text, as follows: ‘[…]  aspects of EOL care until their condition worsened.’

Line 615 – The majority, not majority

We have now corrected this typo.

Line 698 – You don’t need ‘such as’ and ‘e.g.’. ‘such as brain tumors’ would be fine

We have now corrected this typo.

Reviewer 3 Report

Although the subject seems to be relevant, there are a lot of acronyms, references, short phrases and a lot of information compressed in some paragraphs that make difficult to read the paper comfortably. I recommend to reorganize some sections to help the reader to follow/understand the content.

Author Response

Although the subject seems to be relevant, there are a lot of acronyms, references, short phrases and a lot of information compressed in some paragraphs that make difficult to read the paper comfortably. I recommend to reorganize some sections to help the reader to follow/understand the content.

We thank this reviewer for the suggestion. We have now reorganized some sections to help the reader to understand and follow the content, as follows: the abstract has been changed; a relevant keyword has been added; some acronyms have been deleted, and the figure 2 has been moved to supplementary figure 1.

Reviewer 4 Report

This review summarized about advance care planning (ACP) especially in neurodegenerative disorders. This review pointed out some problems including the low frequency of using ACP and the difficulty of deciding the timing to initiate ACP. What do authors think is important to solve these problems? Authors had better mention this point.

Author Response

This review summarized about advance care planning (ACP) especially in neurodegenerative disorders. This review pointed out some problems including the low frequency of using ACP and the difficulty of deciding the timing to initiate ACP. What do authors think is important to solve these problems? Authors had better mention this point.

We thank the reviewer for this comment. We have now slightly changed the Conclusions, as follows: ‘[…] An early initiation of ACP is proposed, also in conditions with a long disease course, such as MS and PD. Training programs for HPs (chiefly physicians and nurses) are key to improve their competences and self-confidence in initiating these conversations in response to a clinical worsening or to a patients’ cue’.

Round 2

Reviewer 3 Report

The paper has been reviewed and improved but the style of english is still, in some paragraphs, fuzzy. Some  phrases must be rewritten  for better understanding.